# Efficiency of a Compressor Heat Pump System in Different Cycle Designs: A Simulation Study for Low-Enthalpy Geothermal Resources

Jakub Szymiczek [1],*, Krzysztof Szczotka [1], Marian Banaś [1] and Przemysław Jura [2]

1   Department of Power Systems and Environmental Protection Facilities, Faculty of Mechanical Engineering and Robotics, AGH University of Science and Technology, Mickiewicza 30, 30-059 Kraków, Poland; szczotka@agh.edu.pl (K.S.); mbanas@agh.edu.pl (M.B.)
2   Polish Chamber of Ecology, Warszawska 3, 40-009 Katowice, Poland; przemyslaw.jura.capital@gmail.com
*   Correspondence: szymiczek@agh.edu.pl

**Abstract:** The development of district heating systems results in a search for alternative heat sources. One of these is low-enthalpy geothermic energy, more available than traditional geothermal energy. However, utilization of these resources is difficult, due to the low quality of the produced heat. To utilize them, the heat pump system can be used. Such a system was designed for this case study of a city in a region of the Polish Lowlands. The data necessary for the design came from the project of the borehole and operational parameters of the existing heating plant. Four heat pump-cycle designs were proposed, modeled, and simulated using Ebsilon software. Afterward, the designs were optimized to achieve maximum coefficient of performance (COP) value. As a result of the simulation, the efficiency of each design was determined and the seasonal COP value was calculated with the annual measured heat demand of the plant. The system based on the cascade design proved the most efficient, with a seasonal COP of 7.19. The seasonal COP for the remaining basic, subcooling, and regenerator variants was 5.61, 3.73, and 5.60, respectively. The annual heat production of the designed system (22,196 MWh) was calculated based on the thermal power of the designed system and historical demand data. This paper presents a simulation methodology for assessment of the efficiency and feasibility of a heat pump system in district heating.

**Keywords:** geothermal energy; low-enthalpy geothermal; renewable energy sources; heat pump; thermodynamic cycle design; Ebsilon; COP

## 1. Introduction

The use of renewable energy resources is a priority in the EU and the world. The depletion of traditional energy resources forces us to search for alternative sources and technologies. In the case of electricity production, the possible alternatives are wind energy, photovoltaics, and (nonrenewable but less impact on the environment) nuclear energy.

However, electricity is not the only form of energy used by the population. According to the authors [1,2], in the typical European household, space heating and domestic hot water production account for 79% of the final energy use. This is a share of energy use that cannot be provided with wind energy or photovoltaics. Nuclear energy can be used for heat as utilization of waste heat [3,4]. However, since heat cannot be efficiently transported over large distances, its use is very limited.

This leads to the conclusion that heating utilization requires different technologies. In district heating, one of the widely used renewables is geothermal energy. It is used in many heating plants in the world, and also in Poland. The main disadvantage of geothermal energy is that its availability strongly depends on locality. A typical doublet exploitation system requires rich aquifers of high-temperature thermal waters [5].

Such systems are widely used in European countries: France, Germany, Italy, and Hungary [6–10].

Heat pump systems make it possible to make use of low-temperature resources. Due to phase transition and change in pressure, the device can transport heat from a lower-temperature source to a higher-temperature sink. With the heat pumps, low-temperature/enthalpy geothermal resources [6,11–14] and even seawater, sewage, or waste heat from data centers [15–19] can be used as a heat source. An example of a large-scale heat pump system used in district heating is Stockholm [13]. The compressor heat pump system with a total thermal output of 180 MW is used with seawater as a heat source. Similar installations are located in Norway [20,21].

To evaluate the heat load of a borehole low-temperature geothermal system, simulation can be performed. In the case of geothermal resources, the actual heat load is dependent on multiple parameters. The simulation of the borehole heat exchange can provide more accurate information on heat load. We [22–24] have researched the simulation of a closed-loop heat exchanger, with additional analysis of the effect of groundwater flow or grout material impact on the heat flow. In the case of well doublets (open-loop system), we [25] performed simulations to decrease land usage of a doublet system. In different research [26], reconstructed geological–hydrogeological models and numerical geothermal simulations were used to define area geothermal potential. In a different study [27], a coupled thermohydromechanical model was developed to help consider the well location to avoid premature thermal breakthrough.

The compressor heat pump analyzed in this study is powered by electricity. To increase its COP value, several factors can be used. The source of mechanical energy in the heat pump is the compressor. Any innovation in its characteristics and construction [28,29] can be used to increase efficiency. Another factor is the refrigerant type. The thermodynamic characteristics of the refrigerant have a key impact on its efficiency [30–32]. However, there are many restrictions caused by the high environmental impact of gases used as refrigerants [33,34]. The factor that will be used in this study is the design of the thermodynamic cycle. The basic heat pump cycle can be modified with additional elements, such as a subcooler, economizer, or regenerator [35,36]. The alteration of the cycle is used to decrease the enthalpy difference between the evaporator and condenser, resulting in decreased compressor work.

The designed cycle's efficiency is dependent on the heat load. To show the results in real-time conditions, the simulation of the parameters based on the historical measured data can be used.

The main aim of this paper was to compare different heat pump cycle designs in a case study of an existing heating plant located in a city in central Poland. A novel modeling approach was used with the simulation of the heat pump system based on measured heating-plant supply data. Ebsilon software, found in power plant or refinery design, was used for this task. The system will use low-enthalpy geothermal resource as a heat source. Simulating energy production with the proposed system according to the parameters of the existing plant was performed in this study. The main findings of this study are a method for simulation of the geothermal heat pump system efficiency based on the parameters of an existing heating plant and a cascade design for boosting the efficiency of the heat pump system.

## 2. Materials and Methods

To create the model of the heat pump system, several assumptions are required. They can be divided into assumptions of the geothermal heat source and heat sink—in this case, the district heating system.

### 2.1. Geothermal Borehole Parameters

The analyzed city is located in central-west Poland (Figure 1), geologically in the region of the Polish Lowlands, known for their low-enthalpy geothermal resources [37–41].

In the vicinity of the city, geothermal resources are used in such places as Poznań and Tarnowo Podgórne [40,41].

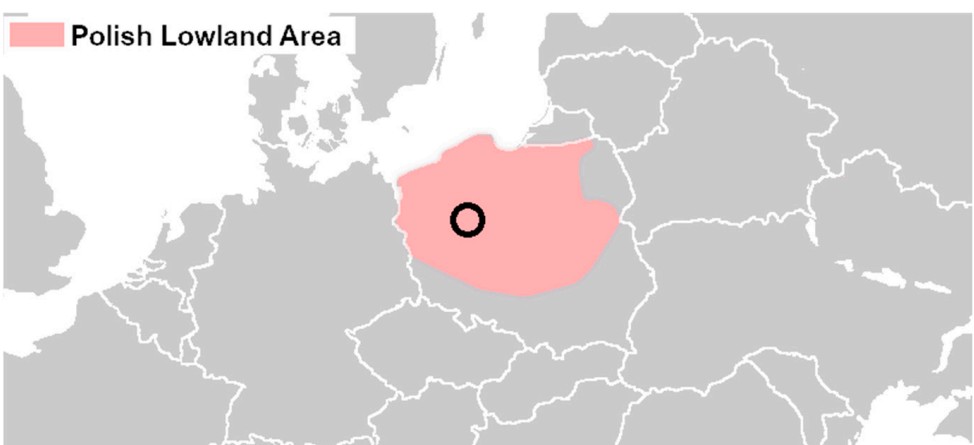

**Figure 1.** Geographic location of the city and the Polish Lowlands in Europe.

The city has been granted a mining permit for the drilling of a borehole and exploitation of the geothermal waters of the Lower Triassic aquifer. The availability of geothermal water in this location was confirmed in the assessment of the geothermal energy resources for the Polish Lowland region [42,43], publications [39,44], and an evaluation of the geothermal energy resources commissioned by the city.

The geological works project for the drilling of the borehole, which was a basis for the concession, anticipates the use of the Lower Triassic aquifer located at approximately 1500 to 1750 mbsl. The expected parameters of the borehole are temperature of the geothermal waters of 65 °C and flow of 85 m$^3$/h. Due to heat loss during the extraction of geothermal waters and possible loss on the surface, the temperature of thermal water at the inlet of the heat pump was determined at 60 °C. These values are later used as an assumption regarding parameters of the geothermal heat source.

According to the project of the borehole, the drilling is to be performed with a normal rotary method using a roller or cutting auger. The borehole diameters of the first four columns are 24″, 18$^5$/$_8$″, 13$^3$/$_8$″, and 9$^5$/$_8$″. The filter column, diameter 148 mm, is to be installed at a depth of 1570–2100 m. This pipe will be suspended on a hanger at a depth of 1570 m and sealed with a pacer.

The expected borehole parameters can be used to calculate thermal power, obtained from the geothermal system, using the following formula [45]:

$$P_{geot} = Q_w \cdot \rho_w \cdot c_w \cdot (T_w - T_i) \tag{1}$$

The $Q_w$ and $T_w$ are thermal water flow and temperature, respectively. Their values were assumed based on the geological data for the location [42,43]. To evaluate the density ($\rho_w$) and specific heat ($c_w$) of geothermal waters, their composition is required. In the case of a planned borehole, the exact composition is still unknown. However, based on the expected mineralization (250 g/dm$^3$) and type of thermal water (Na-Cl) [42], the two parameters were calculated as a NaCl aqueous solution. Based on the mineralization value, the solution mass concentration is 22%, $\rho_w$ = 1148 kg/m$^3$, and $c_w$ = 3354 J/kg·K [46,47]. The $T_i$ is the maximum temperature of reinjected geothermal waters. In this particular case, exploitation of the geothermal aquifers is to be performed with a doublet system. The cooled geothermal water is to be reinjected back into the aquifer to ensure renewal in the aquifer and dispose of highly mineralized water. In the system, the cooling temperature of reinjected waters was set to a minimum of 30 °C. This assumption was set following the borehole project and the mining permit. The resulting maximal heat power is 2.727 MW.

Based on the temperature of the geothermal waters (60 °C) and the minimum temperature of the reinjection (30 °C), the temperature difference in the evaporator would be 30 °C. Similar temperatures can be found in other projects incorporating geothermal heat pumps in Creteil, Fresnes, and Plessis-Robinson [12,47]. In these systems, the temperature difference reaches 37 °C and the reinjection temperature 15 °C.

### 2.2. District Heating Parameters

In the analyzed city, there is an existing district heating and hot water system for 15,000 residents. The current heating plant uses natural gas as a fuel. Its maximal heating power is 19 MW. In 2019, the heating plant produced over 22 MWh of heat. Figure 2 shows daily heat production during 2019.

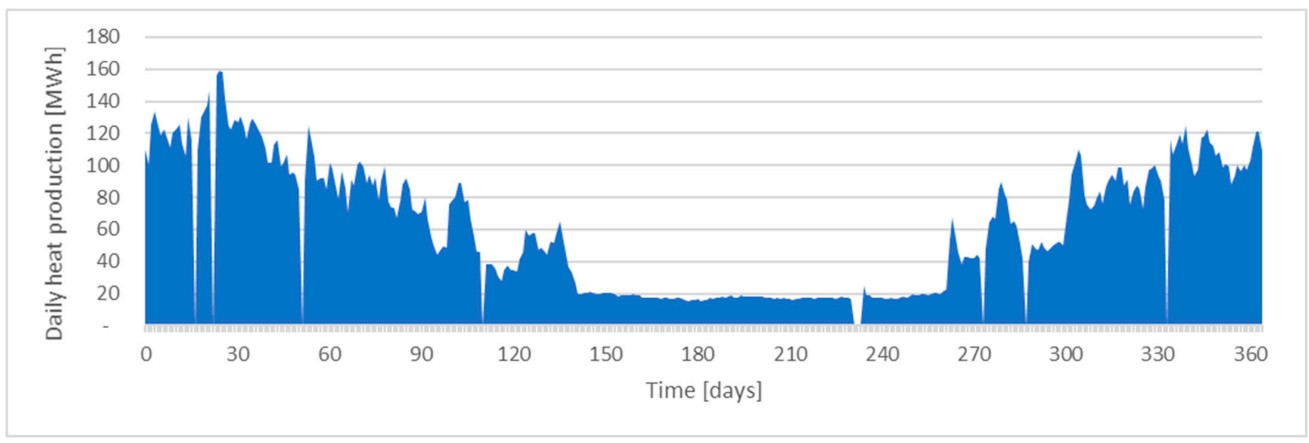

**Figure 2.** Accumulated daily heat production in 2019 in the heating plant.

The data in the graph show the operating time of the heating plant. During the summer season, 140–260 days of the year, the heat demand is the lowest. Heat is used only for the preparation of domestic hot water. For the rest of the year, a clear increasing trend is visible, with peak demand at the beginning and the end of the year. This period is the winter season with the lowest ambient temperatures and highest heat demand. Many fluctuations are visible. During random days, a complete lack of produced energy can be observed. The reason for this is maintenance work or failures that last for over 24 h.

Another important issue is daily variation in heat load. In the analyzed heating plant, no information on daily variation is available. However, the authors [48–50] have noticed this issue in existing heating plants. In [49], the authors found that during the year, the mean daily variation in heat load is 12%. This means that during the day, the heat load can change in the projected span. The highest variation reached 30%, yet it was a single extreme value. Thermal storage can be used to eliminate this problem [48,50]. In the case of the proposed modernization, the variation can be compensated by the heat pumps. During the days when the heat load exceeds the heat pump capacity, the existing gas boilers are to be used. The daily variation can be the cause of a decrease in system efficiency. However, this effect is considered to be insignificant. The overall effect of daily variation is neglected in the simulated system.

The data recorded in the heating plant (Figures 2 and 3) were used to calculate the share of the heat pump system in the energy production and the seasonal value of COP.

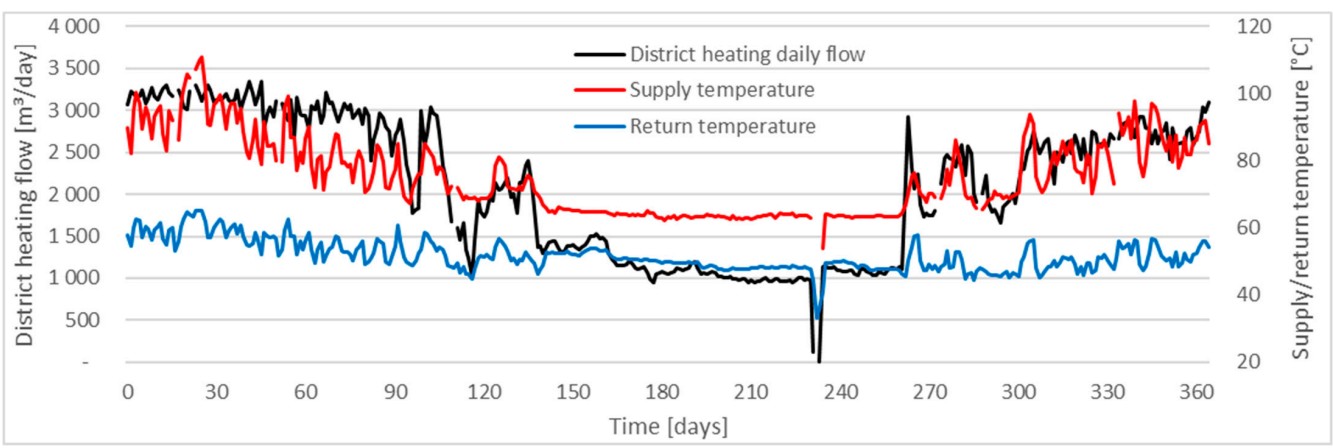

**Figure 3.** Annual data recorded in the heating plant (supply/return temperature, district heating water flow).

The regulation of the output thermal power in the heating plant is performed by adjustment of two parameters: water flow and supply temperature (Figure 4).

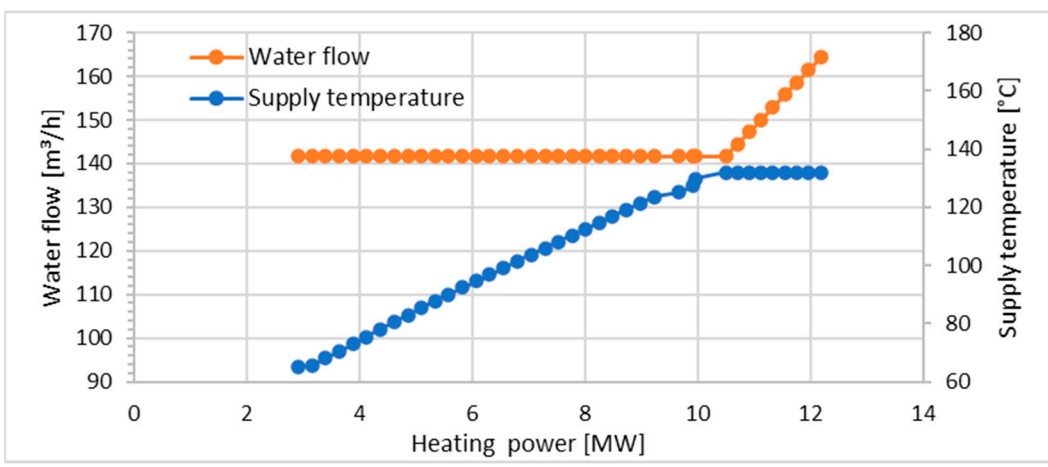

**Figure 4.** Control parameters of the heating plant.

The regulation span is split in two. The first part of the regulation is performed by increasing the supply temperature of the heating water. This leads to a higher temperature difference between supply and return water and higher power yield. The second part of the regulation, for heating power over 10 MW, is performed by increasing the water flow. The second part of the regulation has not occurred during the analyzed period. The demand for over 10 MW corresponds to ambient temperatures lower than −13 °C. Such temperatures are rare in the analyzed region and were not registered during 2019.

The relevant regulation span is performed according to supply water temperature. It has a great impact on heat pump efficiency. The efficiency of the heat pump is expressed as COP. It is dependent on many parameters [51,52], but the one that has the most significant impact is the temperature difference between the heat source (in this case, geothermal waters) and heat sink (district heating water). As we increase the temperature difference, the COP value decreases. This phenomenon has an impact on the actual, final COP value of the system, and is the main aim of the study.

### 2.3. The Proposed Heat Pump System Selection

To evaluate the COP of the compressor heat pump system, modeling and simulation are required. In this study, four heat pump cycles were analyzed to choose the one that

maximizes seasonal COP based on historical data. The refrigerant choice was based on its current broad application in large heat pumps [18,19] and its potential for $CO_2$ emission reduction [31,53] The parameters of the cycle shown in the schemes (Figures 5–8) were obtained with the optimization process for the nominal load, 3.087 MW, and temperature of the district heating return water of 60 °C.

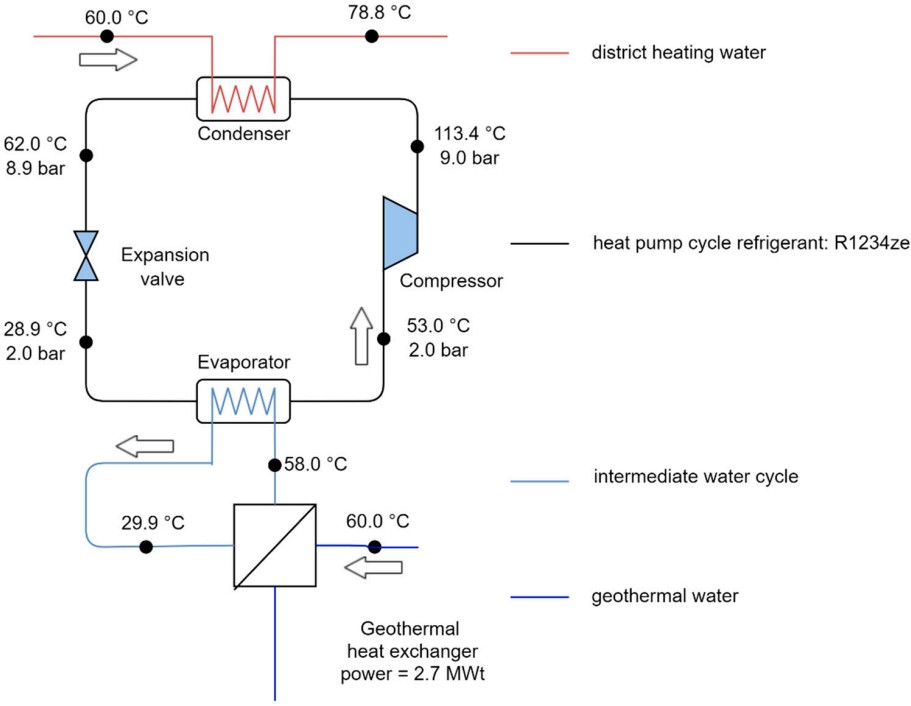

**Figure 5.** Scheme of the basic variant.

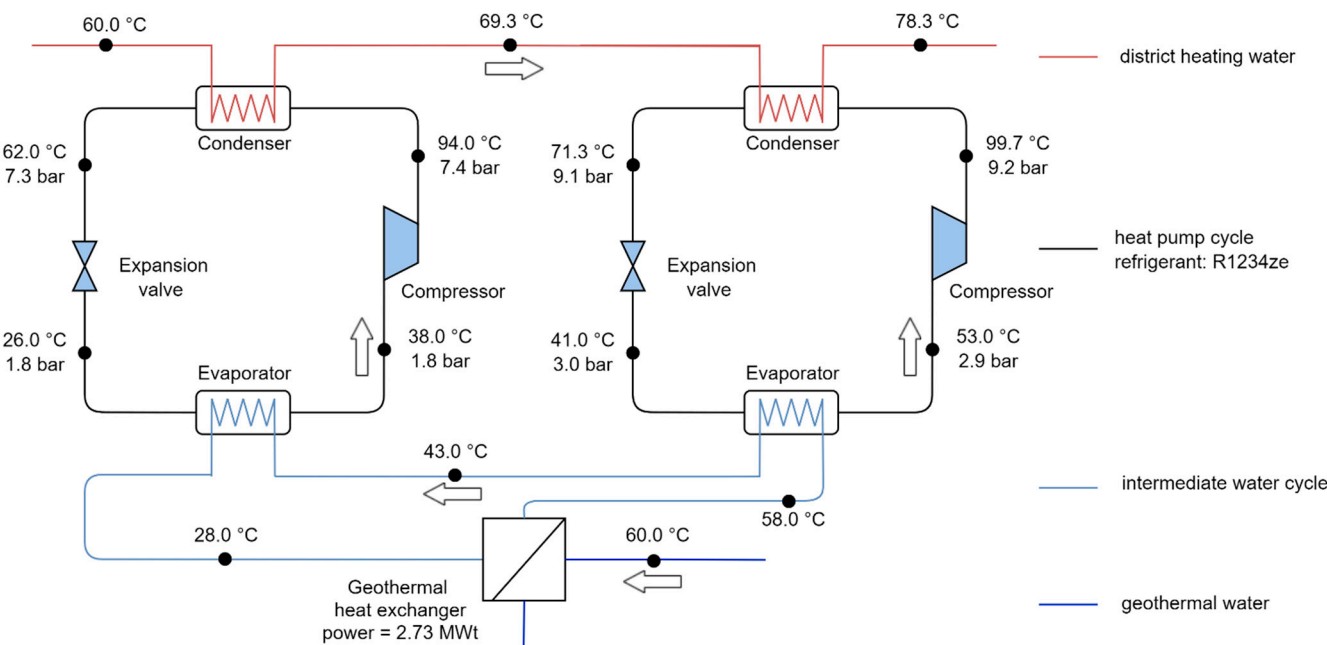

**Figure 6.** Scheme of the cascade variant.

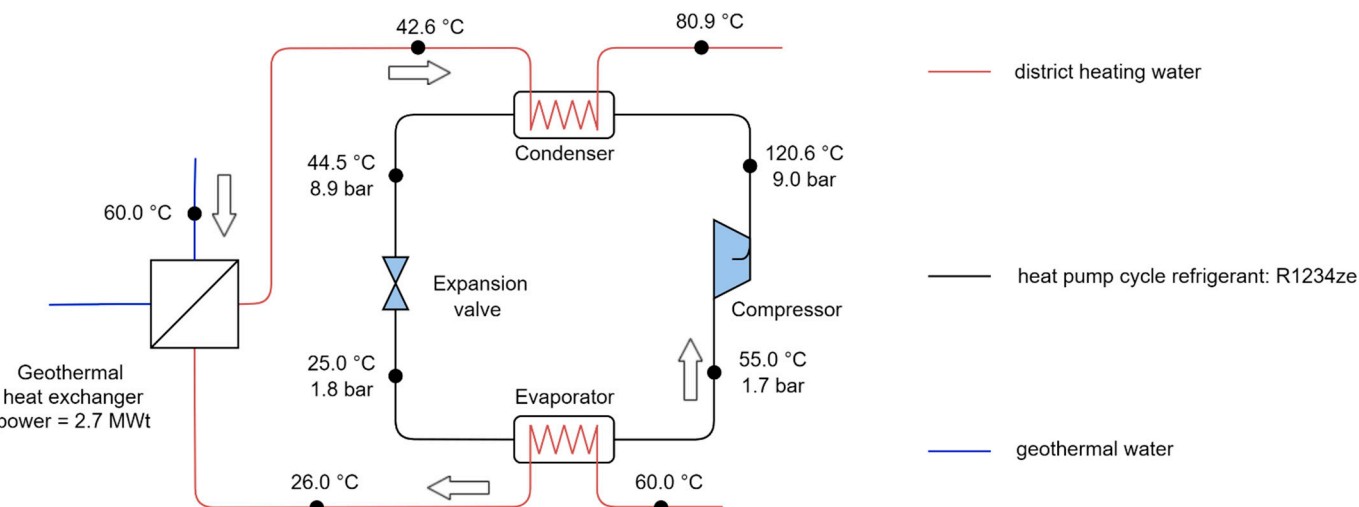

**Figure 7.** Scheme of the subcooling variant.

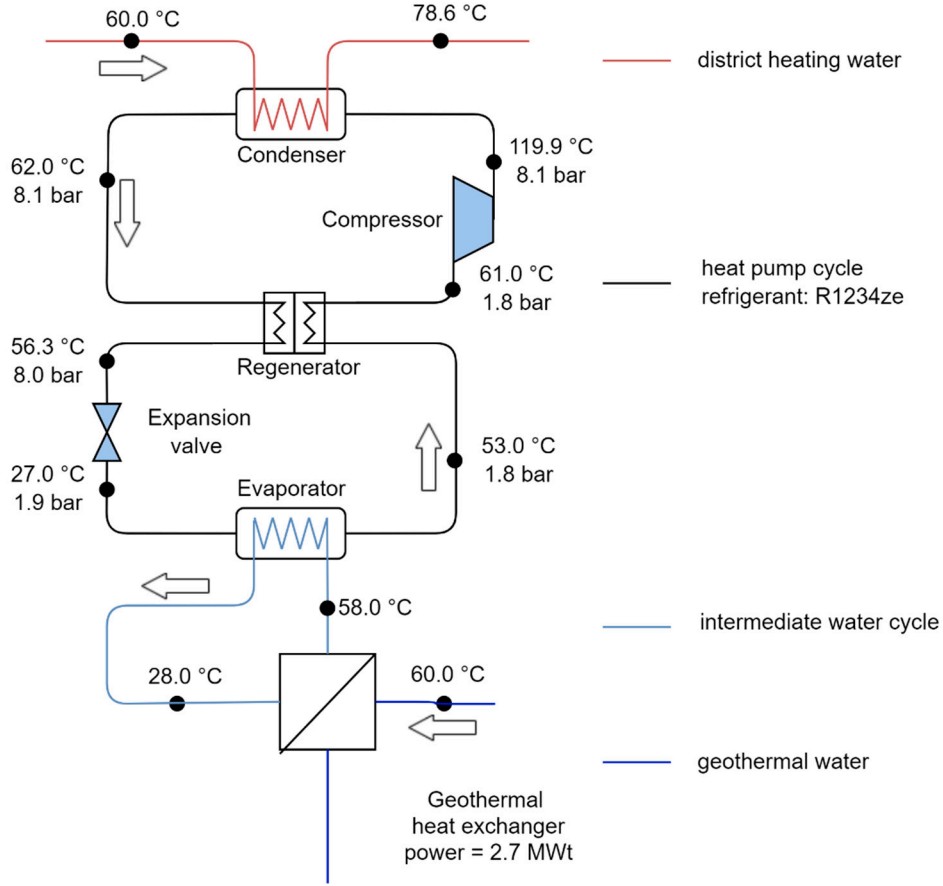

**Figure 8.** Scheme of the regenerator variant.

Different scenarios were analyzed to determine the relation between COP and load. The maximum load was estimated with an assumption that geothermal power of 2.727 MW (calculated in paragraph 2) is entirely used by a heat pump system. The differences in the resulting heat load are an effect of different COP. The compressor work is being transformed into heat.

The variants of the heat pumps are as follows:

*2.4. The Heat Pump System Models*

- Basic variant

The basic variant is a typical heat pump cycle. This scenario was analyzed as a reference scenario to evaluate the gains of more complicated designs. The heat from the geothermal waters is extracted in the evaporator and then transferred to the district heating through a condenser (Figure 5). Temperature and pressure increase is caused by a compressor operation. This variant is the most commonly used in small-scale heat pumps used in single-family housing. It is also used in numerous large-scale installations [18,19].

- Cascade variant

The cascade variant is a more sophisticated solution. The system requires two heat pump units (Figure 6). It was selected based on a similar project: "Le Plessis-Robinson", located in France [54]. Its aim is to increase the efficiency of the system by splitting it into two separate heat pump cycles. Each of the heat pumps is working in the same circuit as the geothermal water and district heating water. However, due to the concurrent alignment of the heat source and the sink, the heat pump works at lower temperature differences. This improvement should increase the COP value of the system. The performance of the cascade system is still a subject of research [32,55–57]. In small heat pumps, such a system is not common, due to the higher initial cost.

- Subcooling variant

This variant (Figure 7) is based on a different principle of heat exchange. The return water from the district is directed to the evaporator. In the evaporator, the heat is withdrawn and the temperature decreased. This cycle design is used in the Ile-de-France region [54], in geothermal and district heating conditions similar to the one analyzed. The district heating water is then directed to the heat exchanger to transfer heat from geothermal water. In the end, district heating water enters the condenser, where it receives the heat previously withdrawn in the evaporator. The heat pump in this system is not used to transfer heat from geothermal water, but to temporarily decrease the temperature of district heating return water. An additional advantage of this variant is the possibility of resigning from the intermediate water circuit. Geothermal water can be used directly. It is not possible in the other alternatives due to technical difficulties in using geothermal water in a heat exchanger with an evaporation process.

- Regenerator variant

The last case (Figure 8) is based on the basic variant with an additional element. The regenerating heat exchanger was added to the refrigerant cycle. The hot side of the heat exchanger is located after the evaporator, and the cold side after the condenser (Figure 8). It aims to transfer the heat leaving the condenser and use it to heat the vaporized working fluid—in short, to regenerate the flow leaving the evaporator. As with previous variants, the regenerator should increase the COP value of the cycle. Such a cycle design is considered an internal heat exchanger [58,59]. It was selected for the analysis due to a simple way of boosting heat pump efficiency.

All of the presented cycle designs can be found in the district heating application. The internal cycle of the refrigerant is similar in all of the variants. The exception is a regenerator, where an additional internal heat exchanger is added to the cycle. The size and dimensions of the device are marginally different. In this case, the exception is a cascade variant, which requires two separate lower-powered devices. A detailed description of each variant can be found next to each of the schematics.

*2.5. Simulations*

The variants were modeled with Ebsilon professional software [60–62]. It allowed the graphical design of the cycle and mathematical modeling of thermodynamics.

The Ebsilon package was chosen due to its thermodynamic simulation capability. This allows for the complete simulation of processes in a heat pump. The package was used to find the best heat pump parameters for the adopted assumptions. The parameters that were the result of simulations were the COP value and heat load of a simulated heat pump cycle. These parameters were later used as data for the evaluation of a designed system.

To perform the simulations, several inputs and assumptions were needed. The inputs were supply temperature and the flow on the side of the heat source (geothermal waters) and the heat sink (district heating water). The discharge temperatures of the heat source and sink water were calculated as an effect of heat pump operation, with an assumption of maximal heat power. In the heat pump cycle, the high/low pressure and refrigerant flow were the controlled parameters. Adjustment of these parameters was the optimization target. The temperatures in the cycle were dependent on flow and pressure values.

To fully simulate the cycle, some additional assumptions were required. To simulate heat exchange in the condenser and evaporator, the terminal temperature difference was set as 2 K to maintain the balance between the best heat exchange and the eventual price of a unit. The pressure drop was assumed at 0.05 bar, a result of a heat exchanger construction. The assumed values were used in previous publications [63,64]. In [65], the authors proved that these values provided the best ratio of effectiveness to price. In numerous publications, the heat loss of a heat exchanger was omitted [32,66,67]. In our work, it was at 5%. The main assumption for the compressor was its isentropic efficiency. Based on the literature [28,29,63,67,68], it was set at 0.75. The characteristic of the compressor efficiency and its dependence on the heat load and refrigerant flow is a default characteristic used by the software. The software default characteristics were also used in heat exchangers. These boundary values were gathered and presented in Table 1.

**Table 1.** Description of all boundary and internal conditions of the modeling.

| Boundary Parameter | Value | Source |
|---|---|---|
| Terminal temperature difference | 2 K | [63–65] |
| Pressure drop | 0.05 bar | [63–65] |
| Heat loss | 5% | [32,66,67] |
| Compressor isentropic efficiency | 0.75 | [28,29,63,67,68] |
| Water pressure | 15 bar | - |

During the simulation, the product of the multiplication of a heat transfer coefficient and a surface area was obtained. This value allowed us to calculate the surface area of a heat exchanger, to make sure that the simulated heat exchangers are feasible for manufacturing.

The calculations were performed for the condenser. The heat transfer coefficient was set following findings presented in [62]. In similar conditions, it was 1–2 $kW/m^2 \cdot K$. Assuming a worst-case value of 1 $kW/m^2 \cdot K$, the calculated surface area of heat exchange of the simulated heat exchangers varies from 269 to 499 $m^2$ in different variants. Such an area can be easily obtained in a commercially available plate heat exchanger.

Figure 9 shows the temperature and heat flow in the simulation of the basic variant evaporator (left) and condenser (right). The hot sides are transferring heat with a positive value, while the cold sides with negative. The figure shows that the simulated heat exchange is feasible.

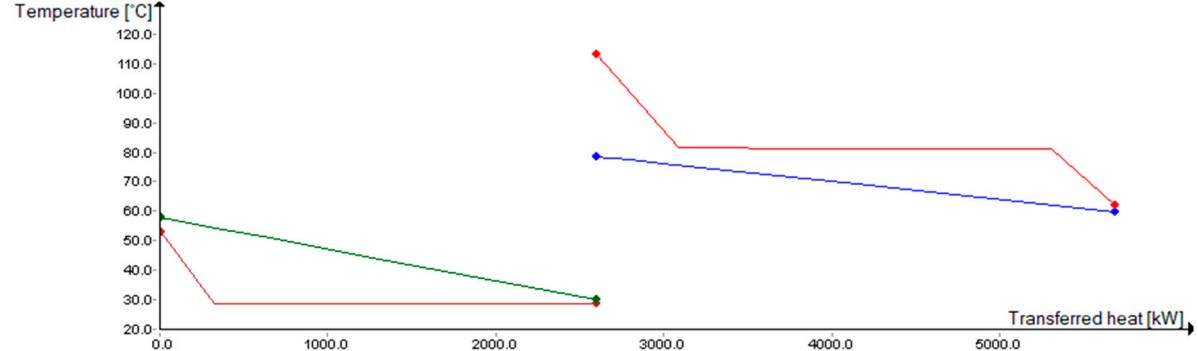

**Figure 9.** Diagram showing the temperature change of the basic variant condenser (**left**) and evaporator (**right**).

The parameters of the cycle were as follows: pressures before and after the compression and flow of the refrigerant. All these values have an impact on the final COP of the cycle. The exemplary model is presented in Figure 10.

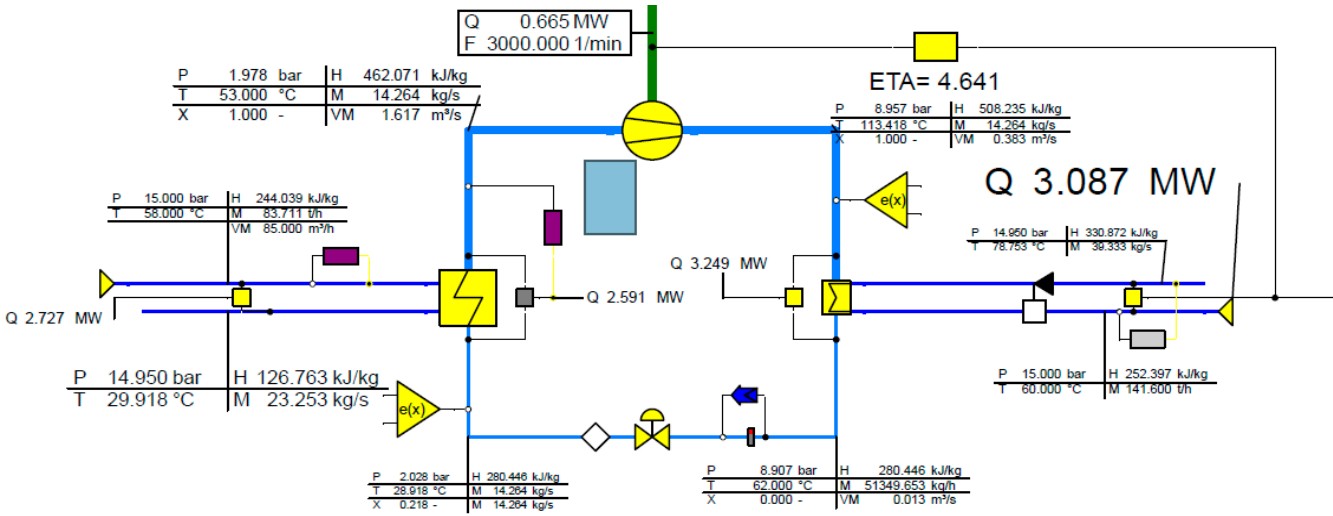

**Figure 10.** View of the Ebsilon professional window—basic variant.

A total of 128 models were calculated for each of the four heat pump alternatives. The operation parameters in each case are presented in Table 2. To maximize the COP value for each of the alternatives, the optimization of pressures and flows was performed. The Ebsilon software has a built-in optimization procedure based on a genetic algorithm [69]. This algorithm is inspired by the natural selection process and uses biologically inspired operators, such as mutation, crossover, and selection [70]. Each variant was optimized using this method.

The maximized function was the COP value, and the input arguments were refrigerant flow and precondenser pressure. The pressure before the evaporator is fixed due to the fixed cooling temperature of the geothermal water. Another restriction in the optimization process was the full evaporation of the refrigerant in the compressor. It was required to ensure that the compressor would be able to work in the designed range of parameters. During the compressor operation, not even the slightest amount of refrigerant can be in a liquid phase, as it could lead to the failure of the device.

**Table 2.** Cycle parameters of all calculated models.

| Variant | Thermal Load [MW] | Supply Temp. [°C] | Return Temp. [°C] | Highest Pressure [bar] | Lowest Pressure [bar] | Max Refrigerant Temperature [°C] | Refrigerant Mass Flow [kg/s] | COP [-] |
|---|---|---|---|---|---|---|---|---|
| Basic | 2.88 | 64.0 | 46.5 | 6.23 | 2.03 | 97.3 | 12.9 | 6.44 |
| | 2.98 | 71.3 | 53.2 | 7.48 | 2.03 | 105.2 | 13.6 | 5.42 |
| | 3.09 | 78.8 | 60.0 | 8.96 | 2.03 | 113.4 | 14.3 | 4.64 |
| | 3.17 | 83.7 | 64.5 | 10.06 | 2.03 | 118.9 | 14.8 | 4.22 |
| | 3.26 | 88.8 | 69.1 | 11.29 | 2.03 | 124.7 | 15.3 | 3.86 |
| Cascade | 2.81 | 63.6 | 46.5 | 6.58 | 2.99 | 84.9 | 7.0 | 8.55 |
| | | | | 5.22 | 1.84 | 78.8 | 6.9 | |
| | 2.90 | 70.9 | 53.2 | 7.78 | 2.99 | 92.2 | 7.4 | 6.86 |
| | | | | 6.23 | 1.84 | 86.3 | 7.3 | |
| | 3.01 | 78.3 | 60.0 | 9.16 | 2.99 | 99.7 | 7.8 | 5.66 |
| | | | | 7.40 | 1.84 | 94.0 | 7.7 | |
| | 3.08 | 83.2 | 64.5 | 10.17 | 2.99 | 104.8 | 8.2 | 5.04 |
| | | | | 8.26 | 1.84 | 99.2 | 8.0 | |
| | 3.17 | 88.3 | 69.1 | 11.28 | 2.99 | 110.1 | 8.5 | 4.50 |
| | | | | 9.21 | 1.84 | 104.6 | 8.3 | |
| Subcooling | 2.98 | 64.7 | 46.5 | 6.01 | 1.77 | 94.1 | 16.0 | 4.78 |
| | 3.15 | 72.4 | 53.2 | 7.31 | 1.77 | 105.8 | 20.9 | 3.45 |
| | 3.43 | 80.9 | 60.0 | 8.98 | 1.77 | 120.6 | 25.5 | 2.58 |
| | 3.69 | 86.9 | 64.5 | 10.34 | 1.77 | 131.7 | 28.5 | 2.19 |
| | 4.06 | 93.7 | 69.1 | 12.03 | 1.77 | 144.8 | 31.5 | 1.89 |
| Regenerator | 2.88 | 64.1 | 46.5 | 5.78 | 1.90 | 97.1 | 13.0 | 6.38 |
| | 2.98 | 71.3 | 53.2 | 6.87 | 1.90 | 107.0 | 13.4 | 5.41 |
| | 3.07 | 78.6 | 60.0 | 8.13 | 1.90 | 119.9 | 13.6 | 4.73 |
| | 3.13 | 83.5 | 64.5 | 9.05 | 1.90 | 128.4 | 13.8 | 4.36 |
| | 3.19 | 88.5 | 69.1 | 10.07 | 1.90 | 137.1 | 14.0 | 4.05 |

## 3. Results

### 3.1. Heat Production

The historical data of heat production in the analyzed thermal plant was used in the simulation of the cumulative daily heat production graph (Figure 11), with a breakdown between the geothermal heat pump system and peak heating, existing gas boiler. In the simulation, the thermal load of the heat pump system is calculated as a 24 h mean operation time providing required historical heating energy. The graph shows that most of the annual heat is produced by the heat pump system.

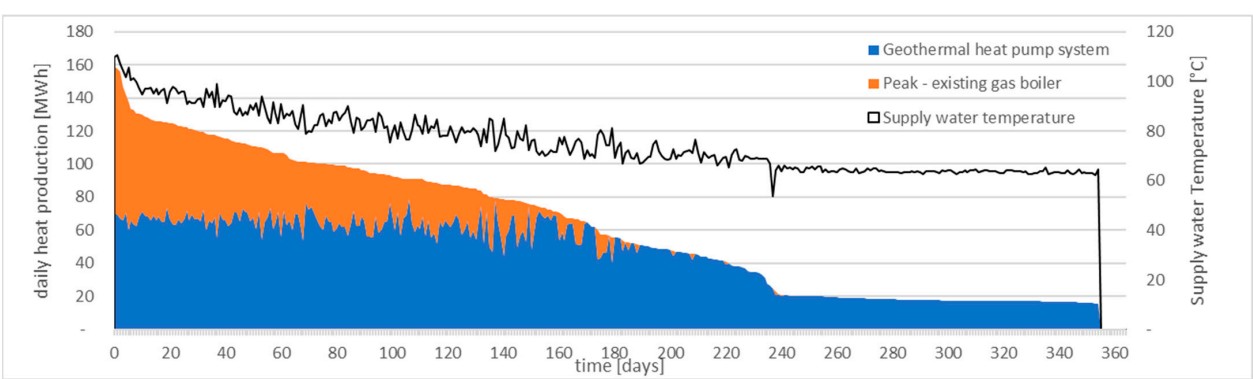

**Figure 11.** Accumulated daily heat production in 2019 with a breakdown of heat sources.

The characteristic "shearing" that can be observed in the graph is a result of a difference between historical data and theoretical regulation of the thermal power. With the increase in daily heat production, the increase in supply and return temperature should be proportional (Figure 4). However, actual measured values differ from this relationship. It may result from fluctuations in the actual heat consumption by final users or the district heating system, and the building's thermal inertia. These phenomena can be the cause of the "shearing."

Another observation is that with increasing heat production, the total energy produced by the heat pumps is decreasing. With increasing heat load, the heat pumps are limited by the heat power of the geothermal source. The heat pumps produce less energy in total share, because of this limitation.

The annual heat production in 2019 was 22,196 MWh. Simulation of the heat production results gives:

- heat produced from geothermal heat pumps = 16,059 MWh
- heat produced from peak gas boiler = 6136 MWh

Exactly 72.4% of the annual energy can be supplied by geothermal heat pumps. This differs from the calculation based on the power of the heat source. Thermal power ordered by the end users in 2019 was 13.305 MW. It is the maximum value of heat provided for district heating during the lowest ambient temperatures. In 2019, the maximum measured power was 7.81 MW. It was obtained by calculating the daily average power level based on the highest daily heat production. The nominal power of the heat pump system is 3.087 MW, only 39% of the maximum measured value and 23% of the ordered power. Even though the power of the heat pump system is much lower than ordered, due to the seasonal distribution of the heat production, it can provide 72.4% of the total annual heat.

### 3.2. COP and SCOP

Another part of the results is related to the efficiency of the heat pump system. The COP value was calculated for each proposed alternative. The value was calculated for the 32 different load parameters (Table 2). In the table, we present only the five, evenly distributed load parameters.

The highest COP was obtained in the cascade system. For every load case, the COP is significantly higher (on average 1.2 times higher) than in the base variant. The second-highest COP can be observed in the regenerator variant. However, the COP difference between the base and regenerator variants does not exceed 4%. The lowest value is found in the subcooling variant.

In the regenerator variant, the COP value was higher when the supply temperature was the highest. In a case where the supply temperature was the lowest, it was lower than the basic. The expected effect of the additional intermediate heat exchanger requires a high temperature difference in the refrigerant cycle. When it is low, the obtained effect is lower than the pressure losses on the additional heat exchanger.

The efficiency value for each variant is inverse to the supply temperature. Its increase leads to a higher difference between the heat source and heat sink, decreasing the system COP value.

To calculate the COP value for the measured historical data, the simulated thermal load had to be approximated with a continuous function. For the calculated values, a polynomial approximation was used to attain the relation of COP and supply temperature (°C; argument "x" in Table 3). Second-degree polynomials were used for this task. The coefficient of determination ($R^2$) of the polynomials was over 0.99 in each variant (Table 3). This value means that 99.5% (0.995 = root of 0.99) of variations in the reference values from measurements can be explained by the proposed model.

**Table 3.** Polynomial equation of COP approximation with supply temperature as an argument.

| Heat Pump Variant | Polynomial Equation for COP | $R^2$ Value |
|---|---|---|
| Base | $COP = 0.0004x^2 - 0.1127x + 12.144$ | 0.9978 |
| Cascade | $COP = 0.0007x^2 - 0.1892x + 17.997$ | 0.9976 |
| Subcooling | $COP = 0.0006x^2 - 0.1602x + 12.614$ | 0.9995 |
| Regenerator | $COP = 0.0004x^2 - 0.1052x + 11.640$ | 0.9774 |

The argument of the function is the value of supply water temperature.

Obtained polynomial equations were used to calculate average daily COP values for the measured data. Further, daily values were used to calculate the annual average value.

$$COP_{av} = \frac{\sum COP_{daily}}{365} \tag{2}$$

The averaged COP gives insight into the heat pump efficiency during the whole year. Still, it does not show variation in energy production. For this purpose, a seasonal COP (SCOP) was used.

$$SCOP = \frac{\sum COP_{daily} \cdot HPheat_{daily}}{heat_{annual}} \tag{3}$$

The SCOP was calculated as a weighted average, where the weight of the COP value was daily heat pump production. The calculated COP and SCOP values are presented in Table 4.

**Table 4.** Results of COP and SCOP calculations based on historical data from measurements.

| Heat Pump Variant | Average COP | Annual SCOP |
|---|---|---|
| Base | 5.82 | **5.61** |
| Cascade | 7.53 | **7.19** |
| Subcooling | 3.99 | **3.73** |
| Regenerator | 5.79 | **5.60** |

Final values are similar to those calculated for set variants. The important observation is the fact that the SCOP value is close to the COP of the low supply temperature. This observation is valid for each of the alternative systems.

The reason for this is the large number of days when the heat pump is the only source of heat. For 200 days during the year (56%), the peak source is not providing any heat (Figure 11), meaning that during that time heat pumps work with the highest COP value. Low COP is reached during the winter season when the supply temperature is the highest. Even though the heat load during that time is the highest, the additional heat is provided by the peak source.

The order of final SCOP results is the same as in the case of specific load calculation (Table 4). The highest value is observed in the cascade system, second for the system with a regenerator, and the lowest for the subcooling variant.

The highest value obtained in the cascade system is consistent with other findings [35] that splitting the cycle creates two different cycles, each working with lower temperature difference. This is the reason for the better efficiency of the system.

The addition of the regenerator to the basic cycle has proven to have a marginally negative effect on efficiency. The reason for this phenomenon is low gains from additional heat exchangers when the supply temperature is low. In such a scenario, the effect of additional pressure losses in the regenerator is higher than gains from its application.

The subcooling variant is characterized by the lowest COP value, due to its inefficiency in the high-temperature difference. This cycle could prove to be more efficient in systems where the temperature difference is lower, e.g., where the geothermal waters would have a higher temperature or the district heating supply temperature would be lower.

## 4. Discussion

The obtained results lead to the main conclusion that the highest COP and efficiency can be obtained by the cascade system. In the simulation, the anticipated efficiency increase caused by a lower temperature difference in the dual heat pump cycle was observed.

Among the other alternatives, the regenerator variant is an easy way to increase typical heat pump efficiency. The add-on of a single component that does not require additional power can cause a positive efficiency effect. In this study, the addition of the regenerator did not result in an overall COP increase. For the highest load, the regenerator slightly increased the COP. However, in the lowest load case, due to the lower temperature difference, the regenerator is obsolete and is only a source of pressure and heat loss.

The important observation noted during the simulation is that the set assumptions do not have an impact on the overall order of the results. Any change in the compressor effectiveness or heat exchanger parameters causes each of the variants' results to proportionally change. This means that the results of the study are representative of a case with similar temperatures of heat sink and source. The assumption that had the largest impact on the results was the compressor efficiency. The effects of the simulation can be compared to the different studies. In the Le Plessis-Robinson Project, the achieved COP is 4.75 in a cascade system working with a heat source temperature of 38/14 °C and heat sink 70/45 °C. The system heating capacity is 6.850 MW and is covering 78% of the annual heat demand. Its COP is lower than in the simulated cascade system, mainly due to lower temperatures of the heat source. Another existing similar system is a subcooling variant [54] with a COP of 4.65 and temperatures of heat source 76/39 °C and heat sink 55/89 °C. The system COP is higher than the simulated one, mainly due to greater heat source temperature.

Another result of the study is the heat pump output power compared to the annual heat production of the plant. The results show that even though the heat power of the designed geothermic plant is much lower than the actual plant, it provides almost all energy production. This conclusion is beneficial for an upgrade of the existing plants. In the situation where the already existing gas power can be used as a peak heat source, a smaller geothermal plant is beneficial. The limitations caused by the geological structure and availability of the geothermal resources make the geothermal plant expensive in terms of the initial cost. The most expensive part is the drilling of the boreholes. Providing 100% of the heat demand with geothermal energy would require another doublet, and thus would double the investment cost. An increased thermal load of the heat pump system would also result in higher heating temperatures (Figure 4), leading to a lower COP, further diminishing the economic feasibility of the system.

In summary, the carbon dioxide emission of the proposed system was calculated and compared to those of the current heating plant (Table 5). The task was performed using historical heat production, simulated heat source breakdown, and calculated SCOP. The emission factors were obtained from the official National Centre for Emission Management reports [71,72] and European Environment Agency data [73].

The emissions in Polish conditions [71,72] are decreased in most of the variants. Only the subcooling variant causes an increase, and is similar to the currently operating gas boilers. It is an effect of the Polish electricity sector, based mainly on coal power plants. The cascade system has the lowest emissions. If the calculation is based on the mean emissions of all European Union countries [73], the emissions are vastly reduced and can yield a 54% to 64% decrease in overall heat production emissions.

**Table 5.** Estimated $CO_2$ emissions of the designed heating plants.

| Existing Gas Boiler | Proposed Modernization Emissions | | |
| --- | --- | --- | --- |
| | Heat Pump Variant | PL Conditions | EU Conditions |
| | [tones/annum] | | |
| 4416 | Base | 3396 | **2014** |
| | Cascade | 2920 | **1841** |
| | Subcooling | 4491 | **2411** |
| | Regenerator | 3401 | **2016** |

The general conclusion is that for the given assumptions of the case study, the best cycle design is the cascade variant, which allows for reaching the SCOP value of 7.19. It is important to highlight that the assumptions and the specific case are representative of most district heating systems [74] and the geothermal resource assumptions also represent many areas with high availability of low-enthalpy geothermal resources [42].

## 5. Conclusions

The main novelty, presented in the paper is a simulation methodology for the assessment of the efficiency and feasibility of a heat pump system in a district heating system. Another innovation is a comparison of the cycle design presented as a case study. This method allows for choosing the most efficient solution at the design stage.

The presented method can be efficiently used in a different case study, with varying internal conditions and heat source/heat sink parameters. The method can be used in the evaluation of a potential heat pump system investment to choose the most fitting cycle design and evaluate the SCOP of the designed system.

However, the issues that were not analyzed can be a setback in the commercial use of such a cycle design. The first of these issues is an increased investment cost of the system. Even though the summarized heat power is the same as in other systems, meaning the components can be smaller and cheaper, still the cascade system requires double all the components. The higher initial cost can be an issue for potential investors. Also, twice the number of components increases the maintenance cost and risk of possible failure. In the large system, the potential profit, both economic and ecological, caused by the increased efficiency surpasses the increased initial costs. In the small system, the final profits might be lower.

Even further possible modification of the cycle could be the add-on of peak gas boiler flue gas heat recovery [75]. The condensing boiler can produce flue gas with a temperature of 40 °C [76]. This flue gas can be used as an additional heat source for the compressor heat pump during peak load. With the possible extension of the heating network, such a modification can be used when the heat load increases.

Future research directions are studies of different cycle designs, including a previously proposed fusion of cascade and regenerator variants. However, such a variant would lead to a great increase in investment cost (dual heat pumps with additional intermediate heat exchangers), while the expected gains are low. Splitting the cycle into cascade variant and intermediate heat exchanger has a similar effect of decreasing the temperature difference between evaporator and condenser; therefore, their cumulative effect on efficiency is expected to be insignificant. Another interesting direction is a case study with lower enthalpy resources, such as lower-temperature geothermal water or even sewage, and surface waters (lakes, rivers). This could increase the potential area of application for the heat pump systems. The next direction is the economic and ecological analysis of the case study, which could provide further data on the reasoning for use of the heat pumps in large-scale installations.

**Author Contributions:** Conceptualization, J.S.; methodology, K.S.; software, J.S.; validation, P.J.; formal analysis, M.B.; investigation, K.S.; resources, J.S.; writing—original draft preparation, J.S.;

writing—review and editing, P.J.; visualization, J.S.; supervision, M.B.; project administration, K.S. All authors have read and agreed to the published version of the manuscript.

**Funding:** This research received no external funding.

**Institutional Review Board Statement:** Not applicable.

**Data Availability Statement:** The data presented in this study are available on request from the corresponding author. The data are not publicly available due to the interests of the data owner.

**Conflicts of Interest:** The authors declare no conflict of interest.

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
