# Peer review of "Efficiency of a Compressor Heat Pump System in Different Cycle Designs: A Simulation Study for Low-Enthalpy Geothermal Resources"

_energies, doi:10.3390/en15155546_

Round 1

Reviewer 1 Report

he manuscript compares different heat pump cycle designs in a case study of an existing system located in a Polish city. A novel modeling approach was well discussed and used to simulate the heat pump system based on experimental heating plant supply data. The manuscript is interesting and well described. It shows this innovative approach that could be helpful for DHC designers. Just some comments for the authors.

The Introduction needs to be improved: a general part (almost one paragraph) describing the numerical modeling used for geothermal systems is needed. Authors can discuss low temperature geothermal systems simulations in general (please see manuscripts below) and same systems but simulated for DHC.

Please read introduce these manuscripts.

-         Hecht-M´endez J, de Paly M, Beck M, Bayer P. Optimization of energy extraction for vertical closed-loop geothermal systems considering groundwater flow. Energy Convers Manag 2013;66:1–10. https://doi.org/10.1016/j.enconman.2012.09.019

-         Matteo Antelmi, Luca Alberti, Sara Barbieri, Sorab Panday, Simulation of thermal perturbation in groundwater caused by Borehole Heat Exchangers using an adapted CLN package of MODFLOW-USG, Journal of Hydrology 596 (2021) 126106, https://doi.org/10.1016/j.jhydrol.2021.126106, 2021.

-         Luca Alberti, Adriana Angelotti, Matteo Antelmi, Ivana La Licata, Borehole Heat Exchangers in aquifers: simulation of the grout material impact, X Convegno dei Giovani Ricercatori di Geologia Applicata 2016, Università di Bologna, 18, 2016, pubblicato su Rendiconti Online Società Geologica Italiana, Vol. 41 (2016), pp. 268-271, doi: 10.3301/ROL.2016.145, 2016

Line 45-48 The sentence is not clear. Heat pump systems works well also when the aquifer temperature is not so higher. Geo-exchange systems (heat pump + borehole heat exchanger) is a worldwide used technology (in many sectors, from building heating to zoo-technical uses). Please read and implement these manuscripts:

-         Islam, M.M., Mun, H.S., Bostami, A.B.M.R., Ahmed, S.T., Park, K.J., Yang, C.J., 2016.Evaluation of a ground source geothermal heat pump to save energy and reduce CO2 and noxious gas emissions in a pig house. Energy Build. 111,446–454

Line 80-99. Could you give more information about the borehole, such as the methofdology of drilling, the diameters, the cost, etc.?

Figure 6 needs to be re-formatted and scaled.

Line 241. Some words or little paragraph with a description of the similarities and differences between the different heat pump cycles may be useful to the reader

Section 2.4 needs to have a schematic description of all boundary and internal condition of the modeling.

Line 366-371. Which is the aim of this analysis? Please highlight it.

The last paragraph Discussion needs to be subdivided into two different paragraphs: one is discussion and is related to a summary of the experiments and modeling; the last one will be the conclusion section where you can highlight the innovative points of your study and how your approach can be extensively applied to different case studies.

Author Response

Review 1

The contents of the Review are pasted in italics.

The manuscript compares different heat pump cycle designs in a case study of an existing system located in a Polish city. A novel modeling approach was well discussed and used to simulate the heat pump system based on experimental heating plant supply data. The manuscript is interesting and well described. It shows this innovative approach that could be helpful for DHC designers. Just some comments for the authors.

The Introduction needs to be improved: a general part (almost one paragraph) describing the numerical modeling used for geothermal systems is needed. Authors can discuss low temperature geothermal systems simulations in general (please see manuscripts below) and same systems but simulated for DHC. Please read introduce these manuscripts (…)

We have made changes in the introduction. We have added a paragraph on numerical modeling for geothermal systems (lines 60-70), and we have added a part on borehole heat exchangers and open loop systems. We have added the suggested manuscripts on BHE [22-24], and some additional on well doublets [25-27].

Line 45-48 The sentence is not clear. Heat pump systems works well also when the aquifer temperature is not so higher. Geo-exchange systems (heat pump + borehole heat exchanger) is a worldwide used technology (in many sectors, from building heating to zoo-technical uses). Please read and implement these manuscripts (…)

We have completely rewritten the sentence (lines 49-51), and we hope that now it clearly explains the need for heat pumps in low-temperature geothermal systems. We have added the suggested manuscript to the analysis of low temperature/enthalpy geothermal resources (line 55)[14].

Line 80-99. Could you give more information about the borehole, such as the methodology of drilling, the diameters, the cost, etc.?

We have added a part on the borehole drilling technology (lines 117-121).

Figure 6 needs to be re-formatted and scaled

We have reformatted all of the schematics (Figures 5-8).

Line 241. Some words or little paragraph with a description of the similarities and differences between the different heat pump cycles may be useful to the reader

We have added a paragraph that summarizes the presented heat pump cycles (lines 265-270)

Section 2.4 needs to have a schematic description of all boundary and internal condition of the modeling.

We have added a summary of all internal conditions in a table (line 302).

Line 366-371. Which is the aim of this analysis? Please highlight it

We have added a sentence emphasizing the purpose of the approximation (lines 403-404).

The last paragraph Discussion needs to be subdivided into two different paragraphs: one is discussion and is related to a summary of the experiments and modeling; the last one will be the conclusion section where you can highlight the innovative points of your study and how your approach can be extensively applied to different case studies.

We have split the discussion into discussion and conclusions sections. As suggested, in the discussion, we have analyzed the results and modeling. In the conclusions, we emphasized the novelty of the paper, possible application areas and future research directions.

We have also fixed some minor flaws. We have fixed the typo on paragraph number (line 272), and a mistake on the pressure drop value presented in the text (line 292).

We wanted to thank you for the generally positive review and hope that our explanations and corrections answer all your questions and uncertainties.

Reviewer 2 Report

The work deals with efficiency of the compressor heat pump system in different cycle. Four different heat pump cycle designs, modeled, and simulated in the Ebsilon

software I evaluate positively. The paper requires a minor revision to meet the requirements of scientific journals having high IF.

1. Please explain clearly what is novelty in this paper.

2. Please explain criteria of the analysis and measurement methods, criteria of choosing measuring parameters.

3. The conclusion section should be added and adapted such that it is totally supported by the obtained results. It is necessary to add the conclusions and add possibility of application.

I recommend this article to publication in Energies after minor revision.

Author Response

Review 2

The contents of the Review are pasted in italics.

The work deals with efficiency of the compressor heat pump system in different cycle. Four different heat pump cycle designs, modeled, and simulated in the Ebsilon software I evaluate positively. The paper requires a minor revision to meet the requirements of scientific journals having high IF.

  1. Please explain clearly what is novelty in this paper.

We have added the conclusions section in which we have written a paragraph on the novelty in the paper (lines 506-509). We have added a sentence emphasizing the novelty at the end of the abstract (lines 24-26).

  1. Please explain criteria of the analysis and measurement methods, criteria of choosing measuring parameters.

In the simulations section, we have added a few sentences on the reason for the choice of the software for modeling, and the need for the simulated parameters in the evaluation of the designed heat pump cycles (lines 275-280).  

  1. The conclusion section should be added and adapted such that it is totally supported by the obtained results. It is necessary to add the conclusions and add possibility of application.

We have changed the last section. We have split it into discussion and conclusions. The discussion section was dedicated to the summary of the simulations and analysis of the obtained results. The conclusions highlight the novelty of the paper, present possibilities of application and future research directions.

I recommend this article to publication in Energies after minor revision.

We wanted to thank you for the generally positive review and hope that our explanations and corrections answer all your questions and uncertainties.

Round 2

Reviewer 1 Report

The authors applied all the suggestions and improved the quality of the manuscript.